# A fluorescent plasmonic biochip assay for multiplex screening of diagnostic serum antibody targets in human Lyme disease

Eunice Chou[1,2], Erica Lasek-Nesselquist[3], Benjamin Taubner[4,5], Arturo Pilar[6], Ernest Guignon[6], William Page[6], Yi-Pin Lin[4,7], Nathaniel C. Cady[1]*

**1** College of Nanoscale Science & Engineering, State University of New York Polytechnic Institute, Albany, New York, United States of America, **2** College of Medicine, State University of New York Downstate Medical Center, Brooklyn, New York, United States of America, **3** Bioinformatics Core, Wadsworth Center, New York State Department of Health, Albany, New York, United States of America, **4** Division of Infectious Diseases, Wadsworth Center, New York State Department of Health, Albany, New York, United States of America, **5** Department of Biomedical Engineering, Mercer University, Macon, Georgia, United States of American, **6** Ciencia, Inc., East Hartford, Connecticut, United States of America, **7** Department of Biomedical Science, State University of New York at Albany, Albany, New York, United States of America

* ncady@sunypoly.edu

**Data Availability Statement:** All relevant data are within the paper and its Supporting Information files.

## Abstract

Lyme disease (LD) diagnosis using the current two-tier algorithm is constrained by low sensitivity for early-stage infection and ambiguity in determining treatment response. We recently developed a protein microarray biochip that measures diagnostic serum antibody targets using grating-coupled fluorescent plasmonics (GC-FP) technology. This strategy requires microliters of blood serum to enable multiplexed biomarker screening on a compact surface and generates quantitative results that can be further processed for diagnostic scoring. The GC-FP biochip was used to detect serum antibodies in patients with active and convalescent LD, as well as various negative controls. We hypothesized that the quantitative, high-sensitivity attributes of the GC-FP approach permit: 1) screening of antibody targets predictive for LD status, and 2) development a diagnostic algorithm that is more sensitive, specific, and informative than the standard ELISA and Western blot assays. Notably, our findings led to a diagnostic algorithm that may be more sensitive than the current standard for detecting early LD, while maintaining 100% specificity. We further show that analysis of relative antibody levels to predict disease status, such as in acute and convalescent stages of infection, is possible with a highly sensitive and quantitative platform like GC-FP. The results from this study add to the urgent conversation regarding better diagnostic strategies and more effective treatment for patients affected by tick-borne disease.

## 1. Introduction

Tick-borne infection with *Borrelia burgdorferi* sensu stricto is the primary cause of Lyme disease (LD) in the United States, where it was recently predicted to affect ~300,000 new people annually [1] and imparts an economic burden of potentially $786 million each year [2]. In

**Funding:** This work was supported by National Science Foundation IOS1755286 (Y.L. and B.T.), National Science Foundation DBI1757170 (B.T.), Department of Defense TB170111 (Y.L. and B.T.), New York State Department of Health Wadsworth Center Start-Up Grant (Y.L. and B.T.), SUNY Polytechnic Faculty Seed Award (N.C.). Funding agencies did not play a role in the study design, data collection and analysis, decision to publish, or preparation of the manuscript. Ciencia, Inc. provided support in the form of salaries for authors A.P., E.G., and W.P., but did not have any additional role in the study design, data collection and analysis, decision to publish, or preparation of the manuscript. The specific roles of these authors are articulated in the 'author contributions' section. The authors have no conflict of interest to declare.

**Competing interests:** Authors A.P., E.G., and W.P. were employed by Ciencia, Inc. during the course of this study. This does not alter our adherence to PLOS ONE policies on sharing data and materials.

Europe and Asia, various spirochetes in the *B. burgdorferi* sensu lato group have been shown to cause LD [3]. The morbid symptoms of late-stage disseminated LD (e.g. joint pain, facial palsy, extreme fatigue, and heart arrythmia) can generally be avoided with prompt diagnosis and treatment [4–8]. One clue that a patient has early localized LD is a characteristic targetoid rash, usually greater than 5 cm in diameter, called erythema migrans (EM). However, EM may be absent or atypical and non-LD rashes may present similarly [9–11]. Serological diagnosis is therefore often made using the standard two-tier test (STTT) algorithm, which entails initial screening for host antibodies against bacterial proteins using the enzyme-linked immunosorbent assay (ELISA or EIA), followed by confirmatory Western blotting to identify specific IgG or IgM against cultured cell lysate [12]. A positive IgM Western blot result (2 out of 3 antibody targets detected: P24, P39, P41) can be used to diagnose LD cases within 30 days of infection and a positive IgG Western blot result (5 out of 10 antibody targets detected: P18, P21, P28, P30, P39, P41, P45, P58, P66, P93) is used in cases of potentially longer infection [13–15].

The STTT has been a useful diagnostic tool since its standardization by the Centers for Disease Control (CDC) in 1995 [13], but critical limitations have become increasingly apparent [16, 17]. These include low sensitivity and low specificity for early disease [18, 19], inability to monitor treatment progress or diagnose re-infection [20], inconsistencies across tests [21–23], and subjective interpretation of Western blot results [18, 24]. Experts agree that new strategies for diagnosing LD are necessary to address these concerns, pointing to modified two-tier algorithms using only ELISAs [23, 25–27], as well as novel assays in various stages of development [28–30]. Indirect serological testing is common because host antibodies against bacterial products are generally more abundant than direct targets, while direct detection of LD remains challenging [20]. Low titers of IgM and IgG Lyme-specific antibodies are present within the first few weeks of infection and increase as the disease progresses [14]. Important considerations when pursuing indirect detection include achieving high analytical sensitivity, as well as being able to distinguish between treated and active disease. A multiplexed and quantitative strategy may be particularly useful for screening diagnostic targets (Table 1), understanding the individualized immune response to LD infection, and capturing the kinetics of this response during the course of disease and treatment [31]. Here we describe a protein microarray, in the form of a compact biochip, that can be analyzed with high sensitivity using grating-coupled fluorescence plasmonics (GC-FP) technology. This strategy uses surface plasmon resonance to enhance the signal of a fluorescent reporter molecule by 100 times [32, 33] and has been used to quantify a variety of targets [34–36], including those of the STTT Western blot [37, 38].

In this study, we systematically optimized and validated the GC-FP biochip for detection of specific serum antibodies in patients with untreated LD, patients treated for LD with antibiotics, and various negative controls using well characterized serum samples [22, 43]. We hypothesized that GC-FP analysis could be used to effectively: 1) screen for antibody targets relevant to various stages of LD, and 2) allow us to develop a new algorithm for diagnosing disease status that is more sensitive, specific, and informative than the STTT. Our efforts involved physical

**Table 1. Current and potential targets for LD serodiagnosis.**

| Diagnostic Markers | Comments | Ref. |
|---|---|---|
| P18, P23 (OspC), P28 (OspD), P30, P39 (BmpA), P41 (FlaB), P45, P58, P66, P93 | Standard 2-tier test Western blot targets | [15] |
| VlsE | Standard 2-tier ELISA target | [39] |
| ErpG, ErpY, ErpL | OspE/F-like proteins | [40] |
| DbpA, DbpB | Decorin-binding proteins | [41] |
| BBA65, BBA69, BBA70, BBA73 | IgM reactivity in early Lyme disease | [42] |

assay development and characterization (i.e. determining the analytical sensitivity, resilience to changes in assay conditions, and variation across replicates), as well as generating a method to analyze the quantitative data (i.e. determining signal detection cut-offs and diagnostic criteria) to provide meaningful outputs with high diagnostic sensitivity and specificity. We show that the GC-FP immunoassay is a reliable and versatile platform that can be used to detect femtomoles of antibodies with only 5 μl of serum. When applied to LD serology, the assay detected IgG and IgM serum antibodies at various stages of disease, including changes in specific antibody levels during convalescence. Notably, we were able to screen a novel set of antigens and design a diagnostic algorithm that may be more sensitive than the STTT for detecting early LD, while maintaining 100% specificity. Findings from this work add to the critical discussion regarding practical replacement of the STTT with a more appropriate test that ultimately promotes better human health via diagnosis and treatment of tick-borne disease.

## 2. Methods and materials

### 2.1 Recombinant antigens and control proteins

Various recombinant *B. burgdorferi* proteins (BmpA, OspD, OspC, DbpA, DbpB, RevA, ErpG, ErpL, ErpY, and VlsE) were produced in *E. coli* as previously described [38]. Purified recombinant proteins, BBA65, BBA69, BBA70, and BBA73 (gift from Dr. Robert Gilmore), as well as P41 and P58 (Surmodics IVD Inc.), were also obtained. SDS-PAGE was used to validate the identity of antigens by the expected molecular weights. Bovine serum albumin (BSA; Sigma-Aldrich) and human IgG (Sigma-Aldrich) were used as negative and positive spotting controls, respectively.

### 2.2 Serum samples

The use of human blood serum samples for this study was approved by the SUNY Polytechnic Institute IRB. Human serum samples were obtained from the Lyme Disease Biobank, Centers for Disease Control, and from Dr. Susan Wong (NY State Department of Health). All samples were de-identified and thus analyzed anonymously. Sample collection by the original collection agencies was performed with written consent. We received patient serum samples accompanied with results from two-tier testing, as well as a detailed clinical history. These sera were derived from patients with early LD (diagnosed by STTT and/or EM rash), disseminated LD, and convalescent stage LD (76 to 99 days following the first dose of antibiotics). Each convalescent sample was derived from the second blood draw of a patient who also submitted an early LD sample. Some of the disseminated LD samples may have originated from patients treated with antibiotics between 1.5 to 157 days prior to serum collection [43]. Negative sera were also obtained from individuals with no relevant symptoms of disease (non-symptomatic), as well as those with non-LD conditions that potentially cause look-alike symptoms (multiple sclerosis or fibromyalgia). All but the convalescent serum samples were used to obtain training set data for generating ROC curves and setting detection cut-offs (Table 2). Seven additional Lyme(+) patient serum samples (from Dr. Susan Wong) were pooled and used as a positive control in preliminary experiments. Lyme(-) control serum (MBL International) was used in some experiments as a negative control.

### 2.3 Biochip preparation and microfluidic processing

Microfluidic biochips were prepared and processed with serum samples as described previously [37] (Fig 1A). Briefly, gold-coated silicon microchips containing a plasmonic diffraction grating were coated with specified proteins using a robotic spotter (ArrayIt; Spotbot II). The negative and positive controls were printed first and last, respectively, along with up to 4

**Table 2. Serum samples analyzed in this study.** The blood serum of 34 individuals were analyzed, including LD patients and negative controls. LD-positive samples were reported to be either in the disseminated, early, or convalescent stages of disease depending on clinical history. Early and convalescent serum pairs from three patients are listed at the bottom of the table. All but the convalescent serum samples were used to train diagnostic algorithms for LD serodiagnosis.

| Sample | Disease Stage | Serum Source |
|---|---|---|
| Biobank #526 | Non-symptomatic | Lyme Disease Biobank |
| Biobank #538 | Non-symptomatic | Lyme Disease Biobank |
| Biobank #610 | Non-symptomatic | Lyme Disease Biobank |
| Biobank #611 | Non-symptomatic | Lyme Disease Biobank |
| Biobank #664 | Non-symptomatic | Lyme Disease Biobank |
| Biobank #674 | Non-symptomatic | Lyme Disease Biobank |
| CDC #A | Non-symptomatic | CDC |
| CDC #B | Non-symptomatic | CDC |
| CDC #C | Fibromyalgia | CDC |
| CDC #D | Multiple Sclerosis | CDC |
| CDC #E | Fibromyalgia | CDC |
| CDC #F | Disseminated LD | CDC |
| CDC #G | Disseminated LD | CDC |
| CDC #H | Disseminated LD | CDC |
| CDC #I | Disseminated LD | CDC |
| Wadsworth #23 | Disseminated LD | NY Dept of Health |
| Wadsworth #24 | Disseminated LD | NY Dept of Health |
| Wadsworth #29 | Disseminated LD | NY Dept of Health |
| Wadsworth #43 | Disseminated LD | NY Dept of Health |
| Wadsworth #44 | Disseminated LD | NY Dept of Health |
| Wadsworth #64 | Early LD | NY Dept of Health |
| Wadsworth #66 | Early LD | NY Dept of Health |
| Biobank #585 | Early LD | Lyme Disease Biobank |
| Biobank #677 | Early LD | Lyme Disease Biobank |
| CDC #J | Early LD | CDC |
| CDC #K | Early LD | CDC |
| CDC #L | Early LD | CDC |
| CDC #M | Early LD | CDC |
| Biobank #640 | Early LD | Lyme Disease Biobank |
| Biobank #681 | Convalescent LD | Lyme Disease Biobank |
| Biobank #663 | Early LD | Lyme Disease Biobank |
| Biobank #682 | Convalescent LD | Lyme Disease Biobank |
| Biobank #673 | Early LD | Lyme Disease Biobank |
| Biobank #688 | Convalescent LD | Lyme Disease Biobank |

replicate spots per antigen. Protein samples were prepared for printing by combining 500 ug/ml of antigen with spotting buffer (ArrayIt) in a 1:1 ratio. Serum samples were diluted using 0.05% PBST at 1:100 X or as specified. Alexa Fluor 647 anti-human IgG (LifeTechnologies) and Alexa Fluor 647 anti-human IgM (LifeTechnologies) were diluted at 1:400X with 0.05% PBST and used as secondary labeling antibodies. Microfluidic processing entailed initial blocking with Superblock (ThermoFisher), followed by addition of diluted serum sample and then secondary antibody, with washes using 0.05% PBST occurring between the latter two steps. In each step, 500 μl of liquid reagent were moved across the flow chamber of the biochip using a syringe pump at 50μl/min.

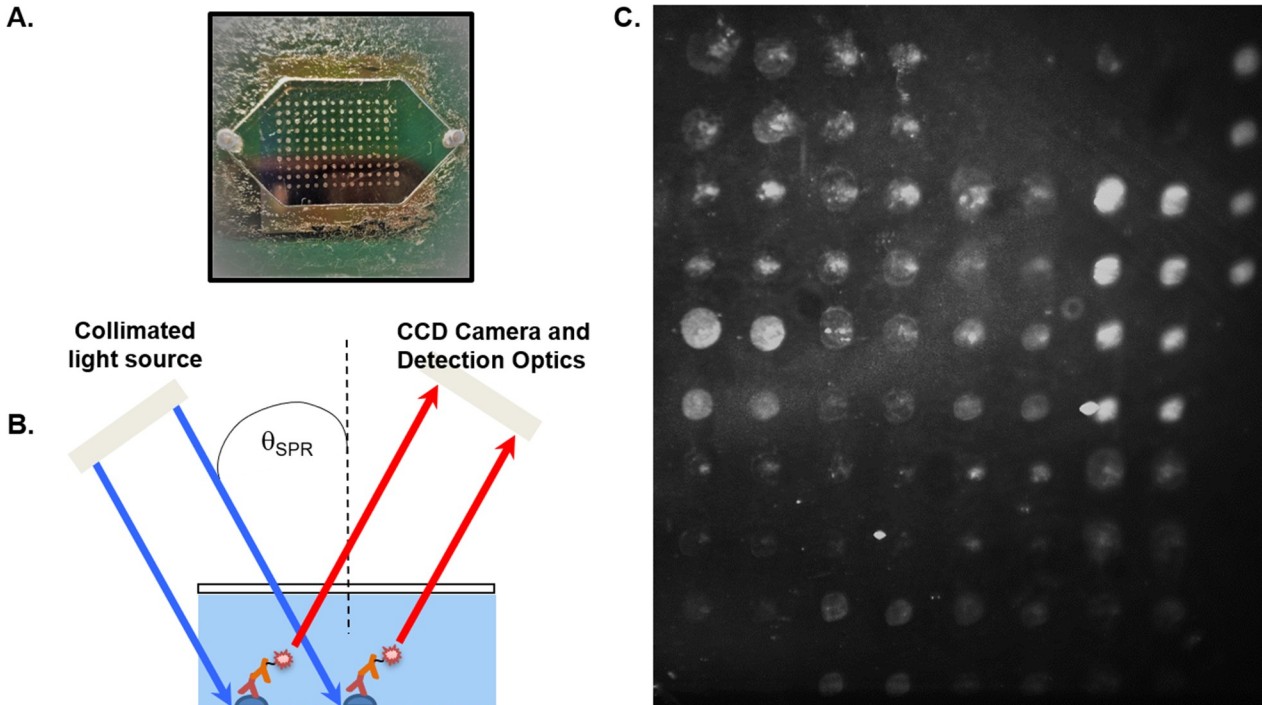

**Fig 1. The GC-FP platform and its linear range of IgG antibody detection.** (A) After being coated with antigens, a GC-FP biochip is assembled with a gasket and window to form a microfluidic chamber, where serum samples and other reagents can be applied. (B) GC-FP analysis on the gold-coated biochip involves using a fluorophore-labelled secondary antibody that couples with the surface plasmon field to emit enhanced fluorescent signal. (C) A representative GC-FP image is shown, containing various LD targets. The fluorescence intensity at each spot ROI corresponds to the amount of detected antibody.

## 2.4 GC-FP image analysis

GC-FP analysis in this study was performed as described previously [37], using a grating-coupled surface plasmon resonance instrument developed by Ciencia, Inc (Fig 1B). The plasmon-enhanced fluorescence signal was measured at 15 seconds exposure time, although other exposure times were used in the event of oversaturation or overall low signal. Validation experiments were conducted using a biochip imaged at 2, 3, 9, and 15 seconds to determine differences in GC-FP signal. A representative GC-FP image of a biochip containing various LD targets is shown in Fig 1C. Protein Array Analyzer for ImageJ [44] was used to set regions of interest (ROIs) and obtain their intensity values. The ROIs for both antigen-coated areas and uncoated background areas were uniformly scaled to accommodate the size of the printed spot. The GC-FP signal (in arbitrary units) was determined by first subtracting the mean background signal composed of ROIs where no protein was deposited on the biochip surface, and then dividing by the mean negative control spot signal consisting of ROIs where BSA was printed onto the biochip. If the average background signal was higher than an antigen ROI signal, causing a negative value after background subtraction, the lowest intensity ROI signal across the surface of the chip was subtracted instead.

## 2.5 Characterizing the sensitivity, flexibility, and reliability of the GC-FP platform

To determine the linear range and analytical sensitivity of the GC-FP platform, a biochip was printed with three replicate spots of human IgG at each concentration: (0.306, 0.638, 1.275,

and 1.785 ng/spot). The biochip was labelled with fluorescent secondary antibody and the average signal intensity for each IgG concentration was determined. The limit of detection and optimal serum dilution was identified by diluting Lyme(+) control serum into three separate samples at 1:1000 X, 1:100 X, and 1:50 X. Samples were flowed from lowest concentration to highest concentration across a biochip containing 16 different *B. burgdorferi* antigens: BBA65, BBA69, BBA70, BBA73, BmpA, DbpA, DbpB, ErpG, ErpL, ErpY, OspC, P41, P45, P58, RevA, and VlsE. After each serum sample, fluorescent secondary antibody was added and the mean GC-FP signal for each antigen was measured.

To determine inter-chip variability, three identical biochips were processed with aliquots from the same patient serum (CDC #H serum sample) to measure specific IgG antibodies with affinity to the 16 antigens. The serum sample was diluted 1:100 X and applied in the microfluidic processing step to obtain GC-FP data for each replicate biochip. A separate negative control biochip was also processed, which contained the same printed antigens, but used the Lyme (-) control sera in the microfluidic processing step. The signal intensities of each biochip were compared to that of the negative control.

## 2.6 Detection of Lyme disease antibodies in patient serum

The GC-FP immunoassay was used to evaluate serum from a patient with early LD (based on clinical symptoms) but a false negative STTT result (CDC #M serum sample). A biochip containing the 16 previously mentioned antigens was processed with this sample and then labelled with fluorescent anti-IgG and anti-IgM secondary antibodies, respectively. A GC-FP image was obtained and analyzed following the addition of each secondary antibody. Thirty-four (34) patient serum samples were processed on biochips printed with 17 antigens (16 mentioned above, along with OspD). These samples underwent labeling with fluorescent anti-IgG only. Acute and convalescent LD serum samples were processed separately in the biochips, but analyzed in pairs, since they originate from two blood draws of each patient. The workflow from printing, to processing, to imaging the biochip is summarized in Fig 2. The binding schemes for detection of IgG only or both IgG and IgM during the microfluidic processing step is also shown.

## 2.7 Constructing a novel diagnostic algorithm

Receiver operating characteristic (ROC) analysis for each antigen was conducted in the ROCR package [45] for R v.3.5.2 (https://www.r-project.org/) using the training set data. Optimal signal cut-offs that maximized sensitivity and specificity were determined for each antigen and only antigens with AUC values $\geq 0.70$ were included in the final diagnostic test. The sensitivity and specificity of every combination of N antigens was evaluated, requiring X (where X = 1 to N) number of antigens to be positive for a sample to be diagnosed as LD. For example, the sensitivity/specificity of every combination of three antigens (N = 3) was evaluated where an LD diagnosis was assigned if X = 1, 2, or 3 antigens were positive. This was performed for up to seven antigens, where it was apparent no increases in sensitivity were gained. Combinations of antigens and scoring criteria (diagnostic algorithms) that yielded the highest sensitivity and specificity were included in the final diagnostic test, where samples that satisfy one or more of these diagnostic algorithms were scored positive for LD.

## 2.8 Statistical analysis

Two-way ANOVA and Fisher's LSD post-hoc analyses were conducted using Prism 6.0 (GraphPad Software). A p-value $< 0.05$ was considered significant.

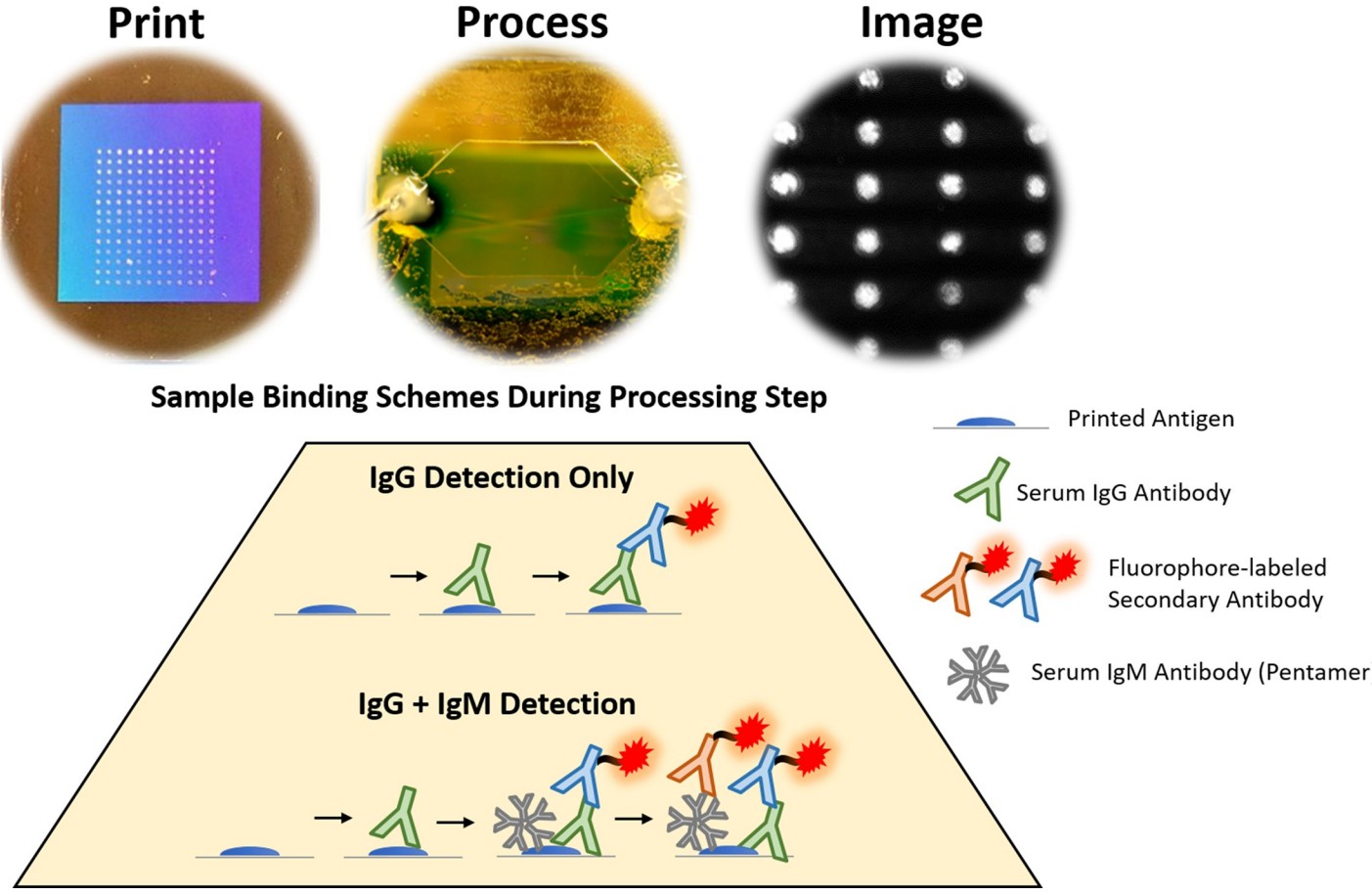

**Fig 2. Summary of the experimental workflow for detecting LD-relevant serum antibodies.** After printing a biochip with antigen spots in the grating area, the biochip is assembled to form a microfluidic chamber. Various reagents are flowed through the biochip during the microfluidic step, in which serum samples are flowed through the biochip. For detection of serum IgG only, a fluorophore-labelled anti-IgG is applied to the chip after the serum sample. For detection of the combined signal of IgG and IgM (secreted in the serum as a pentamer) against each target, fluorophore-labelled anti-IgM is added after the anti-IgG.

## 3. Results

### 3.1 Quantitative analysis of antibody targets

We characterized the quantitative performance of GC-FP for detection of antibodies on gold-coated biochips by determining the linear range of fluorescent signal. The GC-FP signal for each IgG concentration is plotted (Fig 3A). Linear regression analysis of the curve yielded an $R^2$ value of 0.96. Qualitatively, the fluorescent spots were visible at IgG concentrations above 1.275 ng/spot (Fig 3B). Taking into account the reported deposition volume of the microarray pin (5.1 nl), the spot size (0.2 mm radius), and the molecular weight of IgG (150 kg/mol), we estimate that the GC-FP platform may achieve an analytical sensitivity of 8.5 fmol of IgG per 0.13 mm$^2$ spot in a direct antigen-antibody binding context.

We further processed a biochip with increasing concentrations of LD positive control serum. The highest signals were achieved at 1:100 X serum dilution, where 9 out of 16 targets had a GC-FP signal above 10 (S1 Fig). At 1:50 X serum dilution, the GC-FP signal decreased or plateaued.

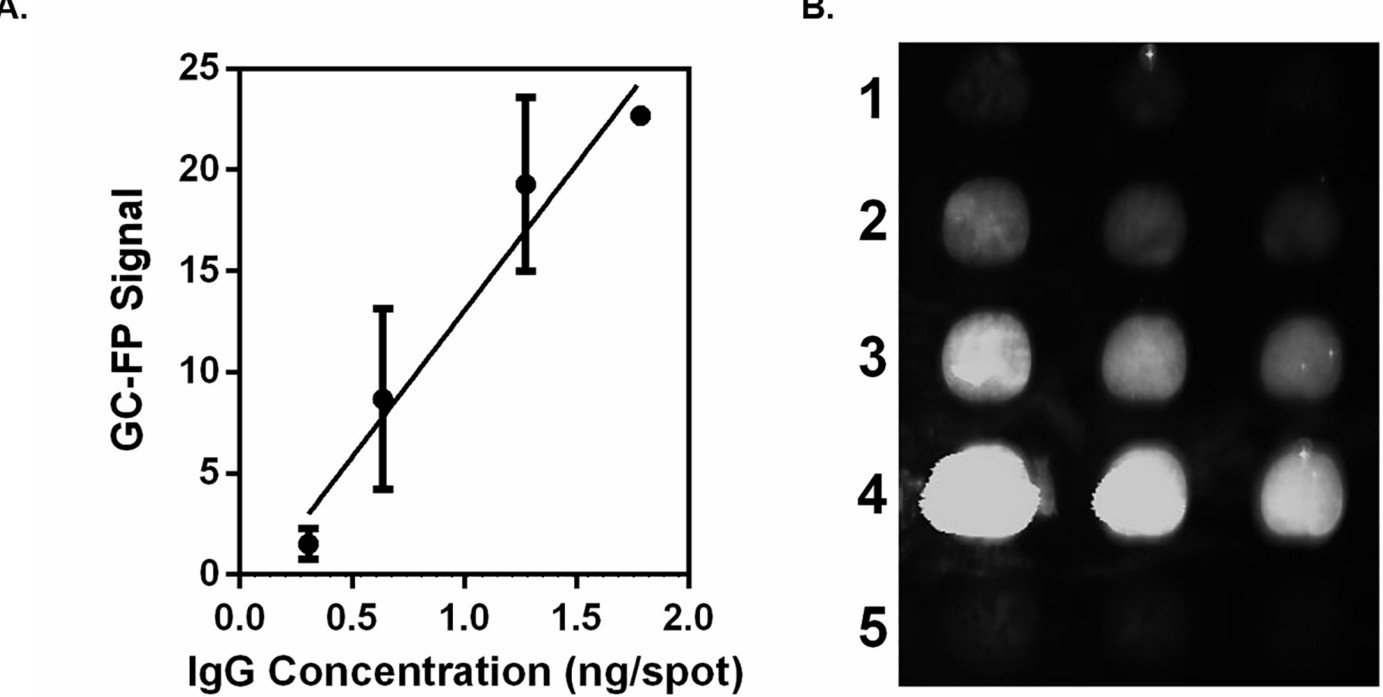

**Fig 3. The GC-FP platform and its linear range of IgG antibody detection.** (A) A GC-FP biochip coated with various amounts of IgG (0.306, 0.638, 1.275, and 1.785 ng/spot) was analyzed. The mean GC-FP signal ± Standard Deviation for each IgG concentration is plotted. Linear regression analysis of the plotted curve yields an $R^2$ value of 0.96. (B) The GC-FP image of spots from lowest to highest IgG concentration (1–4) is shown along with the negative control spots containing 1.275 ng/spot BSA (5). Qualitatively, IgG at 1.275 ng/spot and above is clearly visible above background signal.

### 3.2 Chip stability: Effects of spot size, exposure time, and replicate processing

We investigated the effects of spot size and exposure time on GC-FP results. The signal generated by spots with 400 um diameter were not significantly different from spots with 200 um diameter (t-test; not shown). Moreover, the GC-FP signal of IgG-containing spots at four different exposures times (2, 3, 9, and 15 seconds) were not significantly different from each other (1-way ANOVA; not shown).

The GC-FP signals of three replicate chips were compared with negative control signals (Fig 4). The same ten IgG targets: DbpA, P58, RevA, BmpA, P41, ErpL, BBA69, VlsE, DbpB, and ErpY were found to be significantly higher than that of the negative control for all replicates, although there was inter-chip variation in the mean GC-FP signal intensities.

### 3.3 Detection of low antibody titers in early Lyme disease

GC-FP analysis was used to analyze IgG and IgM antibody binding to *B. burgdorferi* antigens in a patient with early LD. The patient sample scored negative by the STTT diagnostic algorithm (negative EIA result; P41 and P23 detected on the IgM Western blot; P41 and P66 detected on the IgG Western blot). Serum reactivity to each antigen following exposure to fluorescent anti-IgG and anti-IgM secondary antibodies on the biochip is plotted and compared with the results from a negative control sample (Fig 5). After the addition of anti-IgG secondary antibody, 2 out of 16 targets were detected with significantly higher signal than the negative control: BBA65, BBA69. Additional application of anti-IgM secondary antibody led to amplified signal for those two targets, as well as detection of 8 more targets: P58, BmpA, P41, ErpL, VlsE, ErpY, BBA70, and BBA73, or a total of 10 out of 16 detected targets.

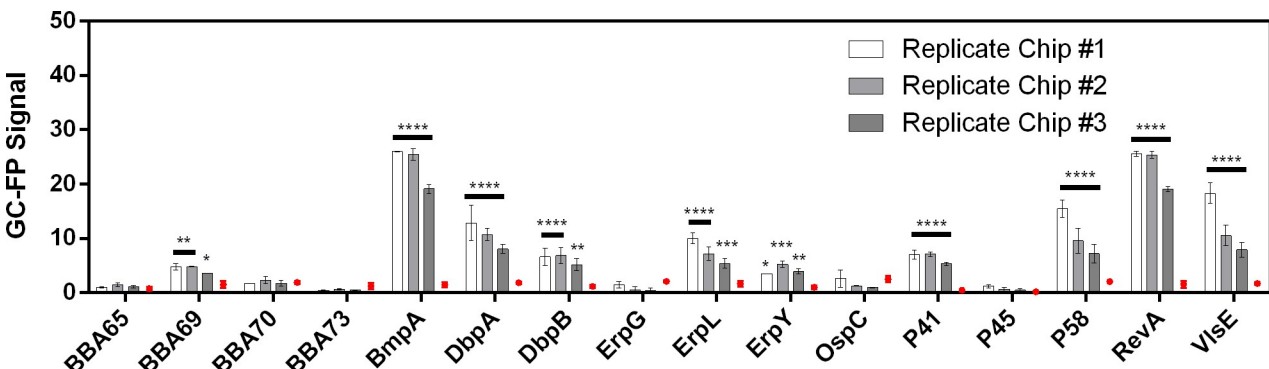

**Fig 4. Inter-chip variation in GC-FP signal.** The mean GC-FP signals ± Standard Error of the Mean is shown for three replicate biochips. The corresponding signals from a biochip processed with negative control serum is also shown (red dot). Black bars indicate where replicate biochips demonstrated significant difference from the negative control, and asterisks indicates a GC-FP signal significantly different from the negative control (****, p<0.0001; ***, p<0.001; **, p<0.01; *, p<0.05; 2-way ANOVA followed by Fisher's LSD test for multiple comparisons).

### 3.4 Developing a GC-FP diagnostic algorithm

The GC-FP data from patient serum samples were used to generate ROC curves for 17 diagnostic targets: BBA65, BBA69, BBA70, BBA73, BmpA, DbpA, DbpB, ErpG, ErpL, ErpY, OspC, OspD, P41, P45, P58, RevA, and VlsE (S2 Fig). Targets that yielded an area under the curve (AUC) ≥ 0.70 included BBA69, BBA70, BmpA, DbpA, DbpB, ErpL, OspC, OspD, P41, P58, and VlsE. We achieved a peak sensitivity of 90% and specificity of 100% with several diagnostic algorithms that scored positive with 2 out of 3, 2 out of 4, and 2 out of 5 targets (Table 3).

We compared the results of our GC-FP diagnostic test to the STTT and IgG Western blot results (Table 4). Two recurring false negative samples (CDC #M and #K) led to 90% (18/20 samples) instead of 100% sensitivity for our diagnostic algorithms. Both samples were derived from patients with early LD who also had a negative STTT result. Of the serum samples tested here, the STTT

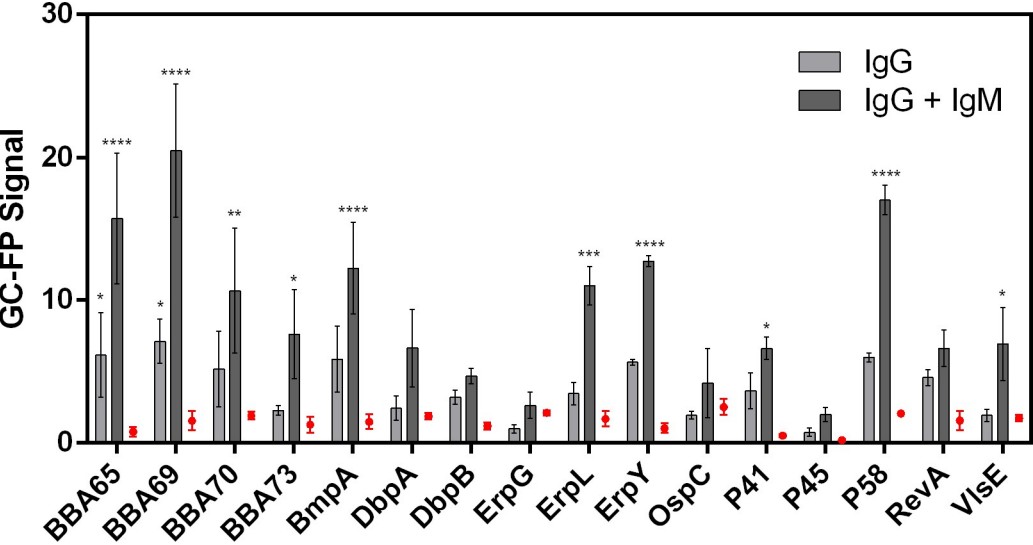

**Fig 5. Detection of specific IgG and IgM antibodies in early LD.** GC-FP was used to evaluate serum from a patient with early LD but a false-negative result based on the STTT. The mean GC-FP signal ± Standard Error of the Mean for IgG detection and additional IgM detection is plotted for 16 antigen targets. IgG against 2 out of 16 targets were detected with significantly higher signal than the negative control: BBA65, BBA69. Additional detection of IgM antibodies led to amplified signal for those two targets and 8 more targets: P58, BmpA, P41, ErpL, VlsE, ErpY, BBA70, and BBA73, or a total of 10 out of 16 detected targets (****, p<0.0001; ***, p<0.001; **, p<0.01; *, p<0.05; 2-way ANOVA followed by Fisher's LSD test for multiple comparisons).

**Table 3. Optimal diagnostic algorithms using combinations of highly predictive antigen targets.** Combinations of 3, 4, or 5 targets, in which detection of at least 2 targets scores positive for the infection, have been found to identify Lyme(+) and Lyme(-) samples with high sensitivity and specificity. The diagnostic algorithms with the highest combined sensitivity and specificity values are listed.

| | Criteria for Positive Diagnostic Score | | |
|---|---|---|---|
| | ≥ 2 of 3 Targets Detected | ≥ 2 of 4 Targets Detected | ≥ 2 of 5 Targets Detected |
| Sensitivity of Diagnostic Algorithm(s): | 90% | 90% | 90% |
| Specificity of Diagnostic Algorithm(s): | 100%% | 100% | 100% |
| | BmpA,DbpB,VlsE | BmpA,DbpB,OspD,VlsE | DbpA,DbpB,ErpL,OspD,VlsE |
| | DbpA,DbpB,VlsE | DbpB,ErpL,OspD,VlsE | BmpA,DbpB,ErpL,OspD,VlsE |
| | | DbpA,DbpB,P41,VlsE | DbpA,DbpB,ErpL,P41,VlsE |
| Groups of Targets with | | BmpA,DbpB,ErpL,VlsE | DbpA,DbpB,ErpL,OspC,VlsE |
| Maximum Sensitivity & Specificity | | DbpA,DbpB,OspC,VlsE | BmpA,DbpB,ErpL,P41,VlsE |
| | | DbpB,ErpL,OspC,VlsE | BmpA,DbpB,ErpL,OspC,VlsE |
| | | DbpA,DbpB,OspD,VlsE | |
| | | BmpA,DbpB,P41,VlsE | |
| | | DbpB,ErpL,P41,VlsE | |
| | | BmpA,DbpB,OspC,VlsE | |
| | | DbpA,DbpB,ErpL,VlsE | |

achieved a sensitivity of 60% (12/20 samples) and specificity of 100%. The sensitivity dropped to 55% (11/20 samples) when only the IgG (not IgM) Western blot was used as the second-tier test.

## 3.5 Changes in serum profile after Lyme disease treatment

GC-FP signal was measured for acute and convalescent LD serum pairs to evaluate changes in specific IgG antibody profile following a standard course of antibiotic treatment. Significant differences between the acute and convalescent groups for each target analyzed are also reported on S1 Table. A high GC-FP signal for targets of the acute sera was frequently paired with a much lower signal for the corresponding convalescent sera (Fig 6). We calculated at least a 5-fold change in GC-FP signal for seven targets in the Biobank #640/681 patient samples (DbpA, OspD, RevA, BmpA, OspC, ErpG, and DbpB), six targets in the Biobank #663/688 group (OspD, BBA73, BmpA, BBA65, BBA69, BBA70), and eleven targets in the Biobank #673/682 group (DbpA, OspD, RevA, BmpA, FlaB, VlsE, OspC, ErpG, DbpB, P58, and BBA70).

## 4. Discussion

### 4.1 Quantitative analysis using the GC-FP biochip is highly sensitive and can determine relative levels of IgG serum antibodies

We implemented a direct binding scheme to investigate the analytical sensitivity and linear range of the GC-FP platform for detecting IgG. This strategy measures on-chip antibody-antigen interactions, as well as the sensitivity of plasmonic fluorescence using minimal assay

**Table 4. Comparison of GC-FP biochip with STTT results.** GC-FP results based on our diagnostic test were compared with STTT results and the standard IgG Western blot results. The total number of serum samples in each group and subgroup is also reported.

| Serum Sample | Lyme Status | STTT | IgG WB | GC-FP |
|---|---|---|---|---|
| CDC #F, CDC #G, CDC #H, CDC #I, Wadsworth #23, Wadsworth #24, Wadsworth #29, Wadsworth #43, Wadsworth #44, Biobank #640, Biobank #673 | + | + | + | + |
| (11) | | | | |
| Biobank #677 | + | + | - | + |
| (1) | | | | |
| – | + | + | - | - |
| (0) | | | | |
| CDC #K, CDC #M | + | - | - | - |
| (2) | | | | |
| Wadsworth #64, Wadsworth #66, Biobank #585, CDC #J, CDC #L, Biobank #663 | + | - | - | + |
| (6) | | | | |
| Biobank #681 | Convalescent | + | + | - |
| (1) | | | | |
| Biobank #682 | Convalescent | - | - | + |
| (1) | | | | |
| Biobank #688 | Convalescent | - | - | - |
| (1) | | | | |
| Biobank #526, Biobank #538, Biobank #610, Biobank #611, Biobank #664, Biobank #674, CDC #A, CDC #B, CDC #C, CDC #D, CDC #E | - | - | - | - |
| (11) | | | | |

**Total Samples: 34** [Lyme (+): 20 / Lyme (-): 11 ]

components. We were able to detect femtomoles of target, which is comparable to the dot-ELISA [46]. A lower analytical sensitivity may be feasible by optimizing the antigen-biochip linkage method to maintain antigen conformation and optimal antibody-antigen interactions [47].

Further application of the GC-FP biochip for multiplexed serum antibody detection involves measuring how each antigen interacts with individual immune responses, which heterogeneously produce antibodies that bind various epitopes on *B. burgdorferi* proteins during infection [48, 49]. Generating a standard curve for absolute antibody quantitation is difficult for a large set of targets and may not provide a satisfactory estimate of an individual's polyclonal antibody profile. Since the diagnostic relevance of the GC-FP immunoassay is to detect antibody levels within a clinical range, we determined the limit of detection by testing various dilutions of positive control serum. By doing so, we evaluated the ability of GC-FP to distinguish between relative concentrations of specific antibodies and determined the minimum serum concentration needed to obtain information about disease status. The findings suggest that a serum dilution of 1:100 X generates the optimal GC-FP signal, where further increases in serum concentration yields marginal improvement and even diminishing returns due to increased background signal. Thus, a major advantage of the GC-FP platform is that we can obtain information about LD status with a small amount of serum (e.g. 5 μl serum needed to generate a 500 μl sample).

## 4.2 The GC-FP immunoassay is resilient to changes in experimental parameters and provides consistent diagnostic results

Practical use of the GC-FP biochip in the clinic depends on system stability, including its resilience to changes in ROI spot size and image exposure. Flexibility in these parameters over the

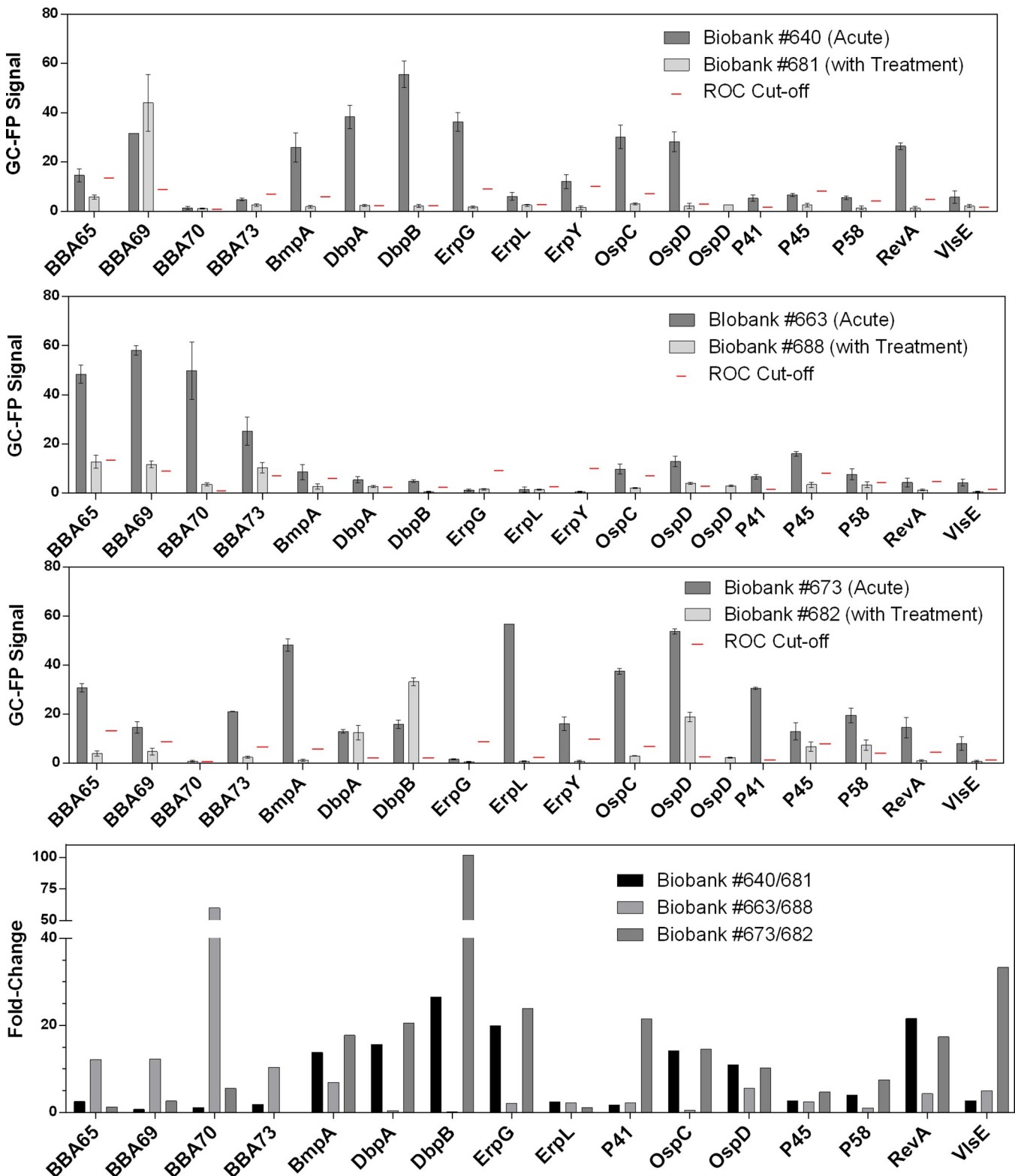

**Fig 6. IgG serum profile following antibiotic treatment.** The mean GC-FP signals ± Standard Error of the Mean for 16 antigens are shown for three acute and convalescent serum pairs. The ROC cut-offs (red lines) provide a reference to determine whether a measured antibody binding response may be considered "positive" for each serum sample. The fold change between acute and convalescent samples are also shown for each serum pair (bottom panel).

course of assay development would allow for a variety of targets and the ability to scale up the number of targets if ROIs are placed in smaller areas and closer together. It is also important to validate the consistency of GC-FP assay results and determine how many replicate chips are required for a reliable result. For three replicate biochips, the identities of targets that generated detectable GC-FP signal (relative to the control) were consistent, although the magnitude of signals varied for individual antigens across replicate chips. Inter-chip variation introduces noise into the data that affects subtle comparisons between signal intensities and may be corrected with optimized assay conditions. For comparisons where the difference in antibody levels is prominent, however, inter-chip variation may be inconsequential. For example, if the final diagnostic result is consistent, one replicate should be sufficient for LD diagnosis.

Some causes for variability may include image processing techniques and protein quality. The GC-FP signal is determined by subtraction of average background signal followed by normalization against the average signal of negative control spots (BSA protein). These steps are thought to account for more drastic differences in overall chip intensity across independent experimental runs, but minor differences across chips may remain and contribute to variation. Inter-chip variability may be further decreased with rigorous control of reagent concentration and quality [50].

### 4.3 GC-FP analysis may have better diagnostic sensitivity than the STTT for early Lyme disease by detecting both IgG and IgM antibodies and incorporating novel disease targets

The ability to detect early LD is a high priority to prevent symptoms of late-stage disease and ultimately promote better outcomes [51]. Since *B. burgdorferi* antigens have been shown to be highly polymorphic [48, 52, 53], LD diagnosis can benefit from multiplexed detection of many different targets to increase sensitivity without losing specificity. Additionally, IgM-based detection of some targets has demonstrated high sensitivity for early LD. These include targets used in the IgM Western blot (OspC, BmpA, P41) [14] and other novel targets (BBA65, BBA69, BBA70, BBA73) [42]. In a patient with early LD and negative STTT result, the GC-FP biochip detected IgG against BBA65 and BBA69, which agrees with a previous study that detected IgM antibodies against these two proteins [42]. The addition of anti-IgM reporter antibody onto the same biochip increased the signal of these targets and led to detection of 8 other targets. These findings suggest potential value in combining IgG and IgM detection for early LD cases. One caveat to sequential detection of multiple isotypes is cross-reactivity of anti-IgM reporter antibody with IgG from patient serum or the anti-IgG reporter antibody. Alternatively, separate biochips can be processed with serum samples blocked to prevent isotype cross-reactivity. Another consideration for including IgM-based detection is the pentamer structure of secreted IgM, which is prone to non-specific interactions and cross-reactivity with extraneous antigens [24]. Hence, IgG-based detection is the preferred strategy for diagnosing LD with high specificity and accuracy. We anticipate that GC-FP detection of IgG against targets that have demonstrated diagnostic potential for early LD, such as OspC and OspF [54, 55], as well as the IR6 portion of VlsE [56, 57], may be useful to include in a multiplexed assay for sensitive and specific diagnosis.

### 4.4 Analysis of GC-FP data can be used to screen for diagnostic targets and lead to new algorithms for determining disease status

In this study, we developed a method to systematically screen biomarkers and determine the most predictive combinations of antigens for use in diagnostic scoring. ROC analysis to establish detection cut-offs was favored over comparing signals for each antigen to that of a pooled

negative control serum. This is because serum reactivity (and cross-reactivity) to LD antigens varied greatly between individuals. Pooled serum controls do not represent any single individual and may underestimate cross-reactivity to some antigens, causing low assay specificity. Moreover, using a negative control reference sample involves statistical analysis with multiple comparisons for each antigen of each sample, which can be cumbersome with increasing sample size.

While testing various combinations of targets, we found that the sensitivity and specificity peaked at 90% and 100%, respectively. This was achieved for diagnostic algorithms that score positive for at least 2 out of 3, 4, or 5 antigens. Sensitivity plateaued and specificity decreased to 91% with combinations of 6 or 7 antigens. Thus, we did not try additional combinations of more than 7 targets. Interestingly, several different combinations of 3, 4, or 5 antigens yielded optimal results. Some antigens were also found in almost all the different optimal combinations (e.g. DbpB and VlsE), which suggests high independent predictability for LD. Additional antigens in the diagnostic algorithm increase sensitivity without decreasing the specificity.

A minimal diagnostic algorithm can decrease economic barriers to scaling up assay production or afford space for detecting additional biomarkers relevant to non-Lyme diseases. However, our analysis of sensitivity and specificity was based on a limited set of serum samples that also constitute the training set. Further analysis using a validation set of samples may allow us to down select from the current set of diagnostic algorithms and determine the minimum number of targets necessary. In this study, samples that satisfied any of the optimal diagnostic algorithms were scored positive for LD. Although multiple algorithms were included in the final diagnostic test, the digital nature of quantitative GC-FP data allowed us to perform the scoring step quickly and generate a definitive output. A similar multi-step strategy could be used to predict disease status in subgroups of patients, where a different set of targets is relevant to each manifestation of disease (e.g. early, late, and convalescent LD, Lyme arthritis, neuroborreliosis).

## 4.5 The potential use of distinguishing between acute and convalescent serum samples in understanding treatment prognosis

Several lines of evidence have pointed to the undulatory nature of serum antibody levels over the course of LD and during treatment [58–61], as well as potentially in post-treatment disease [60, 62–64]. Animal studies have also been used to identify various markers, such as OspA, OspC, OspF, and C6 peptide, that are associated with disease stage and treatment [54, 65]. It should be noted that the presence of antibodies does not equate to active *B. burgdorferi* infection [64], although the antibody profile following treatment has been correlated with symptoms like neuroborreliosis [58] and Lyme arthritis [60]. LD-specific antibody titer generally decreased after treatment and symptom resolution, but for some patients, they remained at levels that were detectable on the Western blot [62, 66]. One study that used automated immunoblotting and software-assisted band analysis identified antibody targets (P28, P30, P31, P34) that were less frequently observed in treated patients without LD symptoms than in treated patients who report persistent LD symptoms [62]. Antibodies against C6 antigen have also been found to decline in patients following treatment for early LD and resolution of symptoms [59]. Thus, changes in the patterns of serum reactivity to antigens may be informative in understanding the disease course and prognosis.

In this study, several targets demonstrated at least a 5-fold decrease in specific IgG titer levels 76 to 99 days following treatment, which mirrors the pattern in acute and convalescent sera of patients treated for acute syphilis [67]. As the standard course of antibiotics for LD usually lasts 10 to 21 days [68], the time frame between the acute and convalescent serum samples in

this study may be large enough to capture disease resolution. In contrast to this, a pattern of increased serum antibodies in the convalescent samples has also been previously shown, which seems to occur when the patient is in the process of mounting a peak immune response to the bacteria [43]. Patients that have been treated during the earliest stages of LD may also never reach a detectable IgG response to the infection [69]. Thus, detailed documentation of the clinical history from tick bite, to symptoms, and to treatment and beyond for each sample is particularly important to extract prognostic information from serum antibody profiles and ultimately benefit the effort to effectively address incomplete treatment, re-infection, post-treatment symptoms.

## 4.6 Conclusions

Recent advances in sensitive molecular detection and high throughput screening of diagnostic targets have brought new hope for improvements in LD diagnostic testing, which has been limited by low sensitivity for early disease and inability to distinguish active versus past infection. The GC-FP immunoassay improves upon current diagnostic methods by affording a larger set of definitive recombinant protein probes with experimental replicates. Thus, we were able to screen for various biomarker targets and generate a diagnostic test that may be more sensitive than the current standard. Moreover, we were able to observe serum profile changes in patients that have been recently treated for Lyme disease. The strategy may eventually enable accurate predictions of disease prognosis in addition to sensitive and specific diagnosis.

## Supporting information

**S1 Fig. Limit of detection for specific LD serum antibodies.** Increasing concentrations of a pooled LD positive control serum (1:1000 X, 1:100 X, and 1:50 X dilutions) were flowed across a biochip spotted with 16 different antigens. The mean normalized GC-FP signal binding of IgG to each antigen is plotted. The highest signals were observed at 1:100 X serum dilution, in which 8 out of 16 targets had signal above 10 (arbitrary units).
(TIF)

**S2 Fig. ROC curves of LD diagnostic targets.** Data from 20 LD-positive and 11 negative control serum samples were included in a training set to generate ROC curves evaluating the independent predictive abilities of 17 potential diagnostic targets: DbpA, P58, RevA, BBA65, BmpA, P41, ErpL, BBA69, VlsE, DbpB, ErpY, BBA70, OspC, P45, ErpG, BBA73, and OspD.
(TIF)

**S1 Table. Significant differences in IgG reactivity following treatment for acute LD.** Acute and convalescent paired samples from the Lyme Disease Biobank was used to evaluate serum profile changes in patients treated for LD. GC-FP analysis was used to compare specific IgG antibody levels for 16 antigens. Significant differences between the acute and convalescent serum pairs for each target are reported (****, $p<0.0001$; ***, $p<0.001$; **, $p<0.01$; *, $p<0.05$; ns, not significant; 2-way ANOVA followed by Fisher's LSD test for multiple comparisons).
(XLSX)

## Acknowledgments

We thank Drs. George Dempsey and Liz Horn for procurement and curation of serum samples from the Lyme Disease Biobank, Drs. Jeannine Petersen and Christopher Sexton for serum samples from the CDC, and Dr. Susan Wong for serum samples from the NY Dept of Health; Ashley Marcinkiewicz for experimental design; Drs. Robert Gilmore and Kevin Brandt

for recombinant protein antigens; Drs. Utpal Pal, John Leong, Chris Li, and Jenifer Coburn for *E. coli* strains to produce recombinant proteins; Wadsworth Center Media & Tissue Culture Core for preparation of *E. coli* and Lyme borreliae culture medium.

## Author Contributions

**Conceptualization:** Eunice Chou, William Page, Nathaniel C. Cady.

**Data curation:** Eunice Chou.

**Formal analysis:** Eunice Chou, Erica Lasek-Nesselquist, Nathaniel C. Cady.

**Funding acquisition:** Yi-Pin Lin, Nathaniel C. Cady.

**Investigation:** Eunice Chou, Benjamin Taubner.

**Methodology:** Eunice Chou, Erica Lasek-Nesselquist, Yi-Pin Lin, Nathaniel C. Cady.

**Resources:** Arturo Pilar, Ernest Guignon, William Page, Yi-Pin Lin.

**Software:** Arturo Pilar, Ernest Guignon.

**Supervision:** Nathaniel C. Cady.

**Visualization:** Eunice Chou, Erica Lasek-Nesselquist.

**Writing – original draft:** Eunice Chou.

**Writing – review & editing:** Eunice Chou, Erica Lasek-Nesselquist, Arturo Pilar, Ernest Guignon, William Page, Yi-Pin Lin, Nathaniel C. Cady.

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
