## [Decision Letter · Decision Letter 0]

12 Dec 2019

PONE-D-19-27828

A fluorescent plasmonic biochip assay for multiplex screening of diagnostic serum antibody targets in human Lyme disease

PLOS ONE

Dear Dr. Nathaniel Cady,

Thank you for submitting your manuscript to PLOS ONE. After careful consideration, we feel that it has merit but does not fully meet PLOS ONE’s publication criteria as it currently stands. Therefore, we invite you to submit a revised version of the manuscript that addresses the points raised during the review process.

We would appreciate receiving your revised manuscript by January 11, 2020 . To enhance the reproducibility of your results, we recommend that if applicable you deposit your laboratory protocols in protocols.io, where a protocol can be assigned its own identifier (DOI) such that it can be cited independently in the future. For instructions see: http://journals.plos.org/plosone/s/submission-guidelines#loc-laboratory-protocols

We look forward to receiving your revised manuscript.

Kind regards,

Sabato D'Auria

Academic Editor

PLOS ONE

Journal Requirements:

1.

2. Thank you for including the following ethics statement on the submission page: 'The use of human blood serum samples for this study has been approved by the SUNY Polytechnic Institute IRB. Human serum samples were obtained from the Lyme Disease Biobank, Centers for Disease Control, and NY State Department of Health. All samples were de-identified and thus analyzed anonymously. Samply collection by the original collection agencies was performed with written consent.'

Please also include this ethics statement in the Methods section of your manuscript.

4. Thank you for stating the following in the Financial Disclosure  section: "This work was supported by National Science Foundation IOS1755286 (Y.L. and B.T.), National Science Foundation DBI1757170 (B.T.), Department of Defense TB170111 (Y.L. and B.T.), New York State Department of Health Wadsworth Center Start-Up Grant (Y.L. and B.T.), SUNY Polytechnic Faculty Seed Award (N.C.). "

We note that one or more of the authors are employed by a commercial company: name of commercial company.

Reviewers' comments:

Reviewer's Responses to Questions

**Comments to the Author**

1. Is the manuscript technically sound, and do the data support the conclusions?

Reviewer #1: Yes

Reviewer #2: Yes

2. Has the statistical analysis been performed appropriately and rigorously? 

Reviewer #1: Yes

Reviewer #2: I Don't Know

3. Have the authors made all data underlying the findings in their manuscript fully available?

Reviewer #1: Yes

Reviewer #2: Yes

4. Is the manuscript presented in an intelligible fashion and written in standard English?

Reviewer #1: Yes

Reviewer #2: Yes

5. Review Comments to the Author

Reviewer #1: The manuscript by Nathaniel C. Cady et al. entitled “A fluorescent plasmonic biochip assay for multiplex screening of diagnostic serum antibody targets in human Lyme disease” describes a new detection method of the human Lyme disease. The aims of this work were to demonstrate that, is possible to detect the human Lyme disease in a different stage of development (early, late/disseminated, convalescent), because the actual gold standard method suffers in the sensibility and/or the specificity. The gold standard method (STTT) is a serological test defined two-tier, so based on two different techniques: indirect ELISA test and confirmatory western-blot. The authors proposed a unique assay based on a grating-coupled fluorescence plasmonic biochip. In brief, is a protein microarray (compact biochip with a gold surface) functionalized with different antigens (multiplexed detection) produced by the B. burgdorferi (the bacteria that cause the Lyme disease).

The authors describe very well how they have found the limit of detection to estimate the analytical sensitivity (8.5 fmol) of the GC-FP platform. Also, they defined the chip stability in terms of replicate processing, exposure time of the measurement and spot size on the chip. The authors defined which antigens are positive detected at each investigated stage of the disease and developing a diagnostic algorithm to evaluate the specificity and the sensibility of the assay. The authors declare that the GC-FP platform achieved 90% of sensitivity and 100% of specificity. In conclusions, the biochip developed to have better diagnostic sensitivity than the STTT for early LD disease, and it is able to distinguish between acute and convalescent patients.

The aim of this work is interesting and the results are promising. The paper is convincing and well done in almost all its parts. Some restructuration of the text may apply to the methods section. In the present form, the paper demands a revision before it can be published, so this reviewer suggests a “minor revision”.

Major issues:

1) This reviewer thinks that could be useful add a subsection in the Methods section that describes better (In details) the assay. The authors reported many references, but an image or a scheme could help the reader to understand better the method proposed and developed.

Reviewer #2: The manuscript by Eunice Chou et al. entitled “A fluorescent plasmonic biochip assay for multiplex screening of diagnostic serum antibody targets in human Lyme disease” describes a protein microarray biochip that measures diagnostic serum antibody targets using grating-coupled fluorescence plasmonics (GC-FP) technology.

The authors conclude that the quantitative, high

sensitivity attributes of the GC-FP approach permit, the screening of antibody targets predictive for LD status, and the development a diagnostic algorithm that is more sensitive, specific, and informative than the

standard ELISA and Western blot assays. The results led to a diagnostic algorithm that may be more sensitive than the current standard for detecting early LD, while maintaining 100% specificity. Moreover the analysis of relative antibody levels to predict disease status, such as in acute and convalescent stages of infection, is possible with a highly sensitive and quantitative platform like GC-FP.

In my opinion the reading of the work is very fluent, the data are well organized. I admit this manuscript, even if I believe that these studies still have points to be strengthened, good results have been achieved.

6. PLOS authors have the option to publish the peer review history of their article (what does this mean?). If published, this will include your full peer review and any attached files.

Reviewer #1: No

Reviewer #2: No

---

## [Author Response · Author response to Decision Letter 0]

3 Jan 2020

Reviewer #1: The manuscript by Nathaniel C. Cady et al. ……..The aim of this work is interesting and the results are promising. The paper is convincing and well done in almost all its parts. Some restructuration of the text may apply to the methods section. In the present form, the paper demands a revision before it can be published, so this reviewer suggests a “minor revision”.

………This reviewer thinks that could be useful add a subsection in the Methods section that describes better (In details) the assay. The authors reported many references, but an image or a scheme could help the reader to understand better the method proposed and developed.

We thank the reviewer for his/her positive comments and have added a figure to the methods section that shows the overall scheme of the biosensing / biosensor platform and the assay that was used for this work. The manuscript text was also modified to describe and refer to the figure. 

Reviewer #2: The manuscript by Eunice Chou et al. ……In my opinion the reading of the work is very fluent, the data are well organized. I admit this manuscript, even if I believe that these studies still have points to be strengthened, good results have been achieved.

We thank the reviewer for his/her positive comments!

---

## [Decision Letter · Decision Letter 1]

24 Jan 2020

A fluorescent plasmonic biochip assay for multiplex screening of diagnostic serum antibody targets in human Lyme disease

PONE-D-19-27828R1

Dear Dr.Nathaniel Cady,

We are pleased to inform you that your manuscript has been judged scientifically suitable for publication and will be formally accepted for publication once it complies with all outstanding technical requirements.

With kind regards,

Sabato D'Auria

Academic Editor

PLOS ONE

Additional Editor Comments (optional):

Reviewers' comments:

Reviewer's Responses to Questions

**Comments to the Author**

1. If the authors have adequately addressed your comments raised in a previous round of review and you feel that this manuscript is now acceptable for publication, you may indicate that here to bypass the “Comments to the Author” section, enter your conflict of interest statement in the “Confidential to Editor” section, and submit your "Accept" recommendation.

Reviewer #1: All comments have been addressed

Reviewer #2: All comments have been addressed

2. Is the manuscript technically sound, and do the data support the conclusions?

Reviewer #1: Yes

Reviewer #2: Yes

3. Has the statistical analysis been performed appropriately and rigorously? 

Reviewer #1: Yes

Reviewer #2: I Don't Know

4. Have the authors made all data underlying the findings in their manuscript fully available?

Reviewer #1: Yes

Reviewer #2: Yes

5. Is the manuscript presented in an intelligible fashion and written in standard English?

Reviewer #1: Yes

Reviewer #2: Yes

6. Review Comments to the Author

Reviewer #1: (No Response)

Reviewer #2: (No Response)

7. PLOS authors have the option to publish the peer review history of their article (what does this mean?). If published, this will include your full peer review and any attached files.

Reviewer #1: No

Reviewer #2: No

---

## [Editor Report · Acceptance letter]

29 Jan 2020

PONE-D-19-27828R1 

A fluorescent plasmonic biochip assay for multiplex screening of diagnostic serum antibody targets in human Lyme disease 

Dear Dr. Cady:

I am pleased to inform you that your manuscript has been deemed suitable for publication in PLOS ONE. Congratulations! Your manuscript is now with our production department. 

With kind regards,

on behalf of

Dr. Sabato D'Auria 

Academic Editor

PLOS ONE